# Membrane palmitoylated protein 2 is a synaptic scaffold protein required for synaptic SK2-containing channel function

Gukhan Kim[1], Rafael Luján[2], Jochen Schwenk[3,4], Melissa H Kelley[1], Carolina Aguado[2], Masahiko Watanabe[5], Bernd Fakler[3,4], James Maylie[6]*, John P Adelman[1]*

[1]Vollum Institute, Oregon Health and Science University, Portland, United States; [2]Instituto de Investigación en Discapacidades Neurológicas, Departamento de Ciencias Médicas, Facultad de Medicina, Universidad de Castilla-La Mancha, Albacete, Spain; [3]Institute of Physiology, University of Freiburg, Freiburg, Germany; [4]Center for Biological Signalling Studies (BIOSS), Freiburg, Germany; [5]Department of Anatomy, Hokkaido University School of Medicine, Sapporo, Japan; [6]Department of Obstetrics and Gynecology, Oregon Health and Science University, Portland, United States

*For correspondence: mayliej@ohsu.edu (JM); adelman@ohsu.edu (JPA)

Competing interests: The authors declare that no competing interests exist.

**Abstract** Mouse CA1 pyramidal neurons express apamin-sensitive SK2-containing channels in the post-synaptic membrane, positioned close to NMDA-type (N-methyl-D-aspartate) glutamate receptors. Activated by synaptically evoked NMDAR-dependent $Ca^{2+}$ influx, the synaptic SK2-containing channels modulate excitatory post-synaptic responses and the induction of synaptic plasticity. In addition, their activity- and protein kinase A-dependent trafficking contributes to expression of long-term potentiation (LTP). We have identified a novel synaptic scaffold, MPP2 (membrane palmitoylated protein 2; p55), a member of the membrane-associated guanylate kinase (MAGUK) family that interacts with SK2-containing channels. MPP2 and SK2 co-immunopurified from mouse brain, and co-immunoprecipitated when they were co-expressed in HEK293 cells. MPP2 is highly expressed in the post-synaptic density of dendritic spines on CA1 pyramidal neurons. Knocking down MPP2 expression selectively abolished the SK2-containing channel contribution to synaptic responses and decreased LTP. Thus, MPP2 is a novel synaptic scaffold that is required for proper synaptic localization and function of SK2-containing channels.

## Introduction

At most excitatory synapses in the central nervous system, such as the Schaffer collateral to CA1 synapses in the *stratum radiatum* of the hippocampus, excitatory neurotransmission is largely mediated by ionotropic AMPA-type (α-Amino-3-hydroxy-5-methyl-4-isoxazolepropionic acid) and NMDA-type glutamate receptors. Yet, an emerging theme is that several conductances that limit membrane depolarization also make substantial contributions to the integrated excitatory post-synaptic potential (EPSP). For example, synaptically evoked $Ca^{2+}$ influx into dendritic spines activates apamin-sensitive SK2-containing channels (small conductance $Ca^{2+}$-activated $K^+$ channels type 2; KCNN2), and their outward $K^+$ conductance shunts the AMPAR-mediated depolarization, effectively reducing the EPSP (*Ngo-Anh et al., 2005*; *Faber et al., 2005*). Kv4.2-containing channels are expressed in spines, close to, but not in the PSD (*Kim et al., 2007*). Synaptic activity evokes $Ca^{2+}$ influx through R-type voltage-gated $Ca^{2+}$ channels in spines that boosts nearby Kv4.2-containing A-type $K^+$ channels to

**eLife digest** The neurons in the brain communicate with each other by releasing chemical messengers across structures called synapses. This signaling always occurs in the same direction: at a given synapse, one neuron sends signals that bind to receptor proteins on the surface of the receiving neuron. Repeatedly signaling across a synapse strengthens it, making it easier to communicate across, and sometimes such stimulation can cause a persistent strengthening of the synapse: this is known as long-term potentiation. Changes in synaptic strength are important for learning and memory.

In the synapses formed between a type of brain cell called CA1 neurons, a protein called SK2 forms part of an ion channel in the membrane of the receiving neuron and is important for synaptic strengthening and long-term potentiation. To work correctly, the SK2 channels must be precisely positioned at the synapse, but the mechanisms responsible for this positioning were not clear.

Now, by experimenting with purified proteins taken from the CA1 neurons of mice, Kim et al. show that SK2 physically interacts with a scaffold protein called MPP2. Further experiments revealed that MPP2 is responsible for positioning SK2 at the synapses, and this allows SK2-containing channels to contribute to long-term potentiation and synaptic strengthening.

During synaptic strengthening, it is possible that SK2 disengages from MPP2, which influences learning. The next step is to understand the processes that dictate this behavior.

further decrease the AMPA-mediated depolarization (*Wang et al., 2014*). In addition, $Ca^{2+}$-activated $Cl^-$ channels are expressed in the spines and provide further inhibitory contributions (*Huang et al., 2012*). Indeed, the sum of these repolarizing conductances may reduce the depolarizing AMPA-NMDA component by more than 50%. It is likely that each of these components can be regulated by a variety of second messenger pathways, greatly expanding the repertoire of targets to fine-tune synaptic transmission. For example, the $Ca^{2+}$ sensitivity of SK2 channels is regulated in an activity-dependent manner by co-assembled protein kinase CK2 and protein phosphatase 2A (*Bildl et al., 2004*; *Allen et al., 2007*) that are engaged by cholinergic signaling (*Giessel and Sabatini, 2010*). Moreover, the various contributions to synaptic responses may be dynamic, changing in response to distinct patterns of activity. Synaptic SK2-containing channels undergo protein kinase A (PKA)-dependent endocytosis upon the induction of LTP by theta burst pairing. The endocytosis of synaptic SK2-containing channels acts together with the PKA-dependent exocytosis of additional GluA1-containing AMPARs to mediate the expression of LTP (*Lin et al., 2008*). Moreover, after the initial expression of LTP and loss of the SK2-containing channel contribution, homeostatic mechanisms act to re-establish the synaptic SK balance (*Lin et al., 2010*). Similarly, Kv4.2-containing channels expressed in spines undergo PKA-dependent endocytosis after the induction of LTP (*Kim et al., 2007*; *Hammond et al., 2008*). Therefore, the appropriate localization, spatial distribution, and orchestrated dynamics of these protein complexes provide a powerful regulator of excitatory neurotransmission and plasticity.

One class of proteins that plays a major role in synaptic organization and dynamics are the MAGUKs (*Elias and Nicoll, 2007*), of which there are 10 subfamilies. These modular, usually multivalent scaffolds bind to synaptic receptors, channels, and signaling molecules to anchor them into their proper locations within the post-synaptic membrane (*Oliva et al., 2012*), creating a spatially and temporally restricted signaling domain (*Hammond et al., 2008*; *Colledge et al., 2000*; *Dell'Acqua et al., 2006*). Thus, within the post-synaptic density of excitatory synapses PSD-95 binds to NMDARs (*Cousins and Stephenson, 2012*), while SAP97 binds to AMPARs (*Howard et al., 2010*; *Leonard et al., 1998*), and Shank and Homer may serve as modular organizers of the lattice of synaptic MAGUKs (*Sheng and Kim, 2000*; *Hayashi et al., 2009*). However, the molecular mechanisms that engender synaptic localization and dynamics to SK2-containing channels are not well understood. There are two major isoforms of SK2 that are expressed in CA1 pyramidal neurons; SK2-L (long) has an extended intracellular N-terminal domain compared to SK2-S (short) and the two isoforms co-assemble into heteromeric channels (*Strassmaier et al., 2005*). In mice that selectively lack SK2-L expression, the SK2-S channels are expressed in the plasma membrane of dendrites and

dendritic spines, yet fail to become incorporated into the post-synaptic membrane. Consequently, the SK2-containing channel contributions to EPSPs and plasticity are absent, and this loss of synaptic SK2-containing channel function enhances hippocampus-dependent learning tasks (*Allen et al., 2011*). To identify proteins that might serve to localize synaptic SK2-containing channels, candidate SK2 interacting proteins were identified. One of them, the MAGUK protein MPP2 (membrane palmitoylated protein 2), is localized to the PSD and is essential for synaptic SK2-containing channel function.

## Results

### MPP2 interacts with SK2

Proteomic analyses (on high-resolution quantitative mass spectrometry) of SK2-containing channels immunoaffinity-purified from rodent whole brain membrane preparations suggested that the MAGUK protein, MPP2 might be an interaction partner. To further investigate this interaction, two newly generated antibodies targeting MPP2 were tested. Probing Western blots of proteins prepared from total mouse brain with either MPP2 antibody detected a predominant band at ~55 kDa. Similarly a single band, ~60 kDa, was detected in proteins prepared from HEK293 cells expressing FLAG-tagged MPP2, but not from mock transfected cells. For both brain and HEK293, pre-incubating the antibodies with the respective immunizing antigen abolished the bands (*Figure 1A,B*).

Therefore, these two antibodies were used for affinity purifications combined with quantitative mass spectrometry (see Materials and methods). The results demonstrated that both antibodies robustly purified MPP2 and co-purified SK2 as well as SK3. In addition, these experiments identified DLG1 (SAP97) and Lin-homologs 7A and 7C as proteins co-assembling with MPP2 in rodent brain (*Figure 1C*; *Supplementary file 1*; *Figure 1—figure supplement 1*; *Figure 1—figure supplement 2*). To rule out off-target effects of the two MPP2 antibodies, we tested their binding of C8-tagged of SAP97, Lin7A, or Lin7C, C8-tagged versions of each of these proteins after expression in HEK293 cells (*Figure 1D*). While all 8-tagged proteins were recognized by anti-C8 antibody only C8-MPP2 was recognized by anti-MPP2 antibodies (*Figure 1D*). MPP2, a member of the p55 Stardust family of MAGUK proteins, is predicted to be a 552 amino acid protein. Like most MAGUKs, MPP2 is a modular protein comprised of several distinct protein-protein interaction domains. There are two predicted L27 domains, followed by a single predicted PDZ domain, and then the SH3-HOOK-GK domains.

To test for direct interaction with SK2 channels, SK2-L and SK2-S, the SK2 isoforms that contribute to synaptic SK2-containing channels (*Allen et al., 2011*), and either C8-MPP2 or PSD-95 were co-expressed in HEK293 cells. Immunoprecipitations were performed using an anti-SK2 antibody raised in guinea pig that is directed to the common C-terminal domain of the two SK2 isoforms, or using IgG as a control. Precipitated proteins were prepared as Western blots. Probing with anti-SK2 antibody that was raised in rabbits and directed against the same C-terminal sequence, demonstrated equivalent SK2 expression for input and after immunoprecipitation in each sample (*Figure 2—figure supplement 1*). Probing with anti-C8 antibody detected a band of the appropriate apparent molecular weight for C8-MPP2 in the sample co-expressed with SK2-S plus SK2-L but not in the IgG control sample (*Figure 2A*). Probing with anti-PSD-95 antibody did not detect co-immunoprecipitation with SK2 (*Figure 2B*). The results demonstrated that C8-MPP2 but not PSD-95 was specifically co-precipitated with SK2. To examine the possibility that SAP97 interacts with SK2-S, C8-SAP97 was co-expressed in HEK293 cells together with either myc-SK2-S or GluA1, a known SAP97 interaction partner (*Leonard et al., 1998*). Immunoprecipitations were performed using anti-SK2 antibody, anti-GluA1 antibody, or IgG as a control. Western blotting with anti-C8 antibody revealed that C8-SAP97 only co-immunoprecipitated when co-expressed with GluA1, despite equivalent levels of input and immunoprecipitated proteins (*Figure 2—figure supplement 1*). To test if MPP2 interacts with the unique N-terminal domain of SK2-L that is required for synaptic localization GST pull-down experiments were performed. While PDZ interactions are important for many MAGUK-partner protein interactions, the unique N-terminal domain of SK2-L does not contain a PDZ ligand motif and using the PDZ domain from MPP2 failed to show an interaction with the N-terminal domain of SK2-L. Therefore, the SH3-HOOK-GK domain of MPP2 was employed. This domain of MPP2 as well the SH3-HOOK-GK domains from SAP97, a MAGUK scaffold of a distinct subfamily

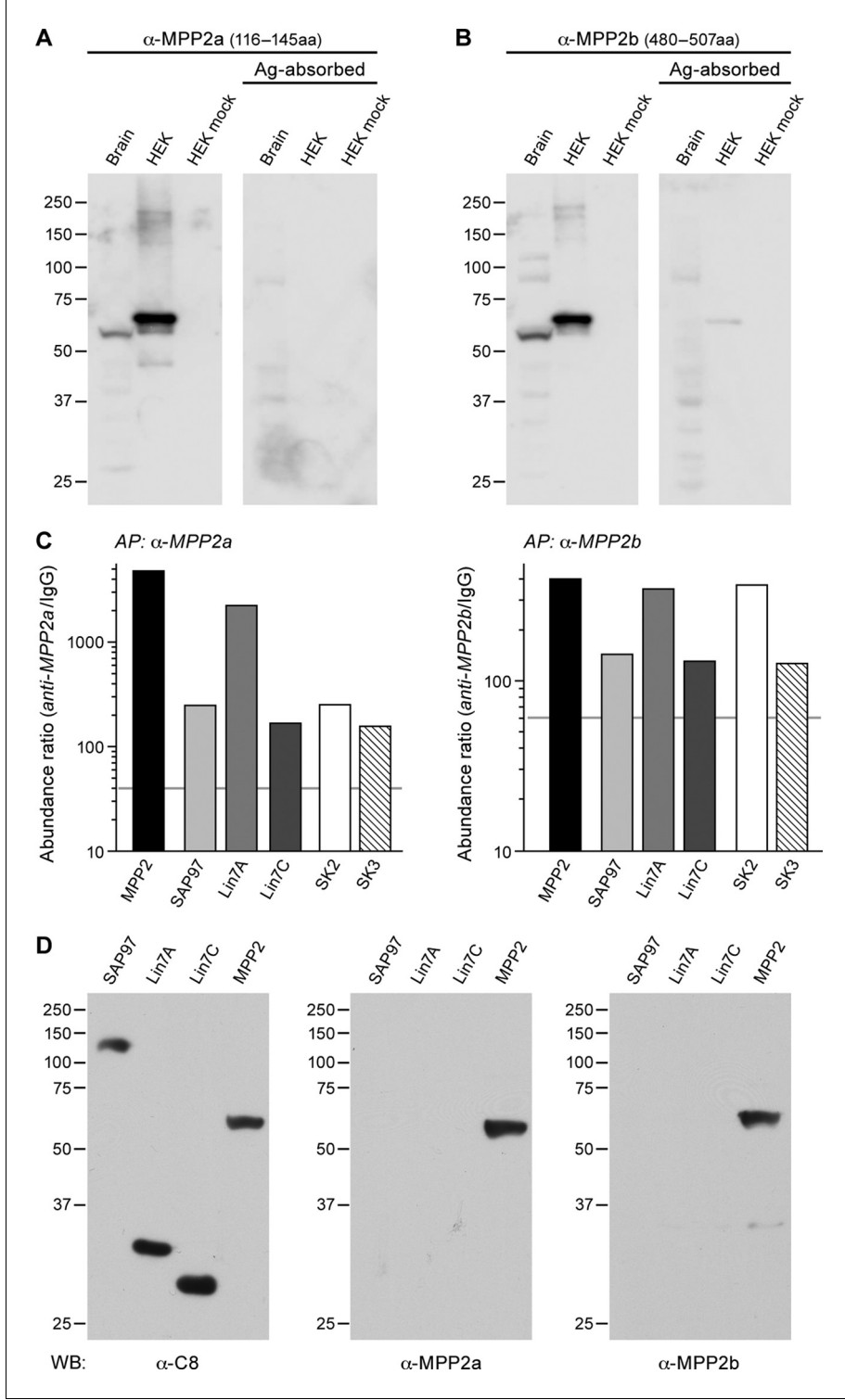

**Figure 1.** MPP2 interacts with SK2. (**A, B**) Western blots were prepared using mouse brain homogenate (Brain; 100 μg) or HEK293 cell extracts (1% of 10 -cm plate), either transfected to express FLAG-MPP2 (HEK) or empty plasmid (HEK mock). Duplicate lanes were prepared and one set was probed with the indicated MPP2 antibody (left panels) while the second set was probed with the same antibody after pre-absorbing with the immunizing antigen (right panels). Native MPP2 and transfected FLAG-MPP2 were detected only in the left panels. (**C**) Bar graphs illustrating abundance ratios determined for the indicated proteins in APs with two anti-MPP2 antibodies and IgG as a negative control. Horizontal lines denote threshold for specificity of co-purification. (**D**) Western blots of

*Figure 1 continued on next page*

*Figure 1 continued*

proteins prepared from HEK293 cells transfected with C8-tagged SAP97, Lin7A, Lin7C, and MPP2, and probed with anti-C8 antibody (left), anti-MPP2a antibody (middle), and anti-MPP2b antibody (right). The MPP2 antibodies recognized only MPP2.

The following figure supplements are available for figure 1:

**Figure supplement 1.** Coverage of the primary sequences of all proteins shown in *Figure 1C*.

**Figure supplement 2.** Coverage of the primary sequences of all proteins shown in *Figure 1C* (continued).

from MPP2 (*Oliva et al., 2012*), or the SH3-HOOK-GK domain from CaCNB4, a non-canonical MAGUK protein that is a beta subunit for voltage-gated $Ca^{2+}$ channels (*Van Petegem et al., 2004*) were prepared as GST-fusion proteins, bound to glutathione-agarose beads, and used as baits for the prey, the His-tagged N-terminal domain of SK2-L. A GST-fusion protein of the C-terminal domain of Kv1.4, and a His-fusion protein of PSD-93 (Chapsyn) were also prepared as positive interaction controls (*Lunn et al., 2007*). Coomassie staining and Western blotting with anti-GST antibody verified equivalent amounts of the baits, either as input or bound to beads (*Figure 2—figure supplement 2*). After exposure to the baits, bound prey proteins were eluted and prepared as Western blots. Probing with an anti-His antibody showed that the SH3-HOOK-GK domain from MPP2, but not from SAP97 or CaCNB4, specifically pulled down the N-terminal domain of SK2-L. As expected,

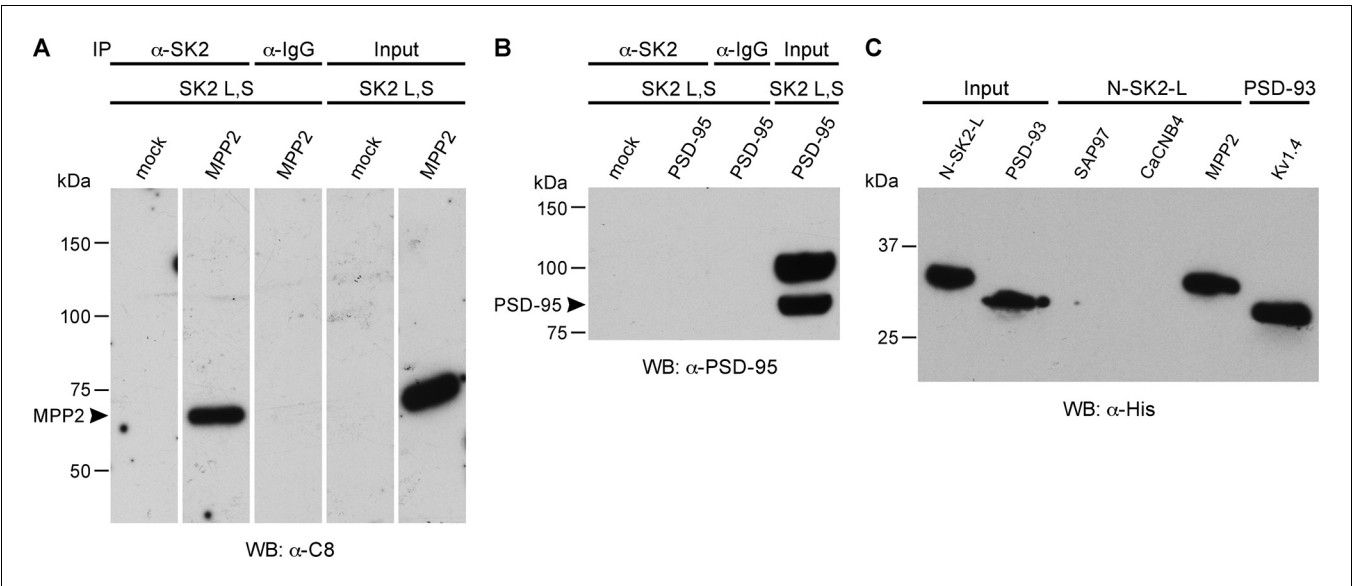

**Figure 2.** MPP2 interacts with the N-terminal domain of SK2-L. (**A**) Co-immunoprecipitations. Western blots prepared from HEK cell lysates expressing SK2-S and SK2-L, either alone (mock) or together with C8-MPP2, immunoprecipitated with either anti-SK2 antibody or IgG (control), and probed for C8-MPP2. Adjacent blot shows input of C8-MPP2. (**B**) A similar experiment except using PSD-95 instead of C8-MPP2. PSD-95 is expressed but does not co-IP with SK2. (**C**) GST-pull-downs. Blot probed with anti-His antibody shows input prey proteins, His-SK2-L N-terminal domain and His-PSD-93. After exposure to GST-baits representing SH3-HOOK-GK domains of SAP97, CaCNB4, or MPP2, the His-SK2-L N-terminal domain was specifically retained by GST-MPP2. Positive control shows interaction between GST-C-terminal domain of Kv1.4 and His-PSD-93. Co-immunoprecipitations and GST-pull-downs were performed in triplicate.

The following figure supplements are available for figure 2:

**Figure supplement 1.** Co-immunoprecipitations.

**Figure supplement 2.** GST-fusion protein expression.

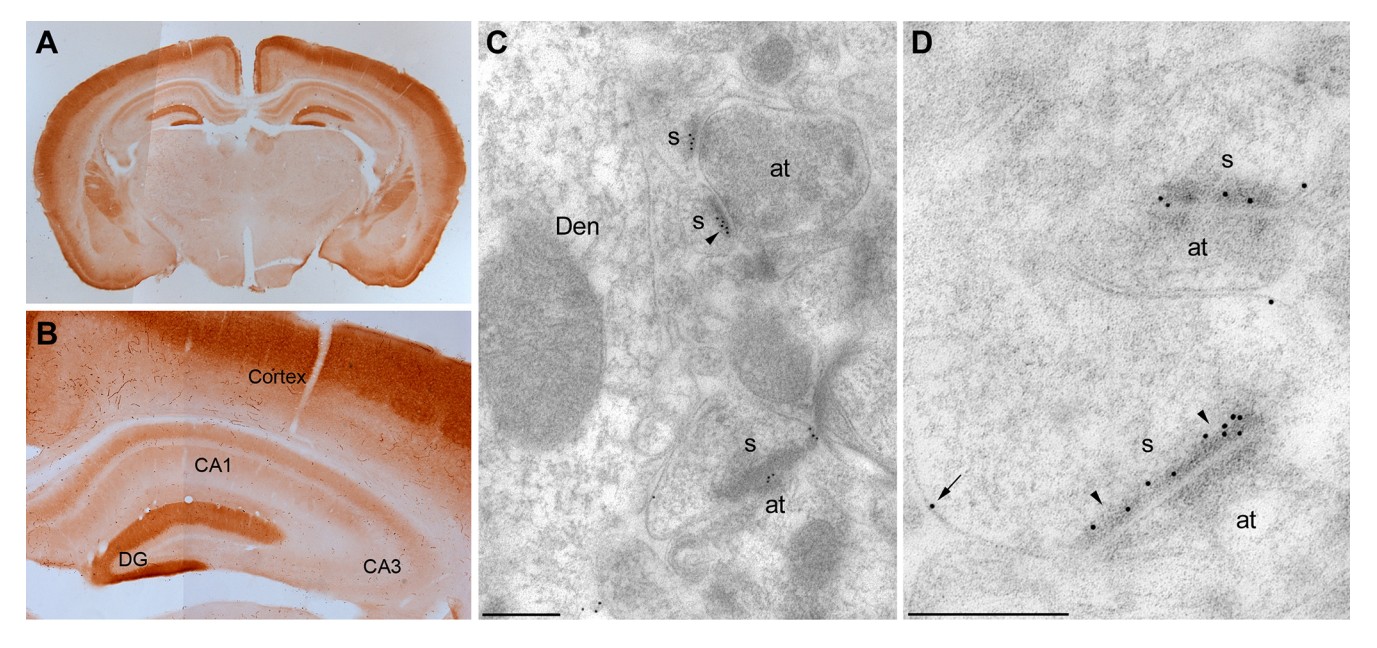

**Figure 3.** Localization of MPP2 in the hippocampus. (**A,B**) Light microscopic images of anti-MPP2 antibody labelling in hippocampus. MPP2 was present throughout the hippocampus and was prominent in the dendritic arbors of area CA1. (**C,D**) Electron micrographs of the hippocampus showing immunoparticles for MPP2 in the *stratum radiatum* of the CA1 region of the hippocampus, as detected using a post-embedding immunogold method. Immunoparticles for MPP2 were detected along the PSD (arrowheads) of dendritic spines (s) of CA1 pyramidal cells establishing asymmetrical synapses with axon terminal (at), as well as at extrasynaptic sites (arrows) of dendritic spines (s). Den, dendritic shaft. Scale bars in **C,D**: 0.2 μm.

the C-terminal domain of Kv1.4 pulled down PSD-93 (*Figure 2C*). Taken together these results suggest that MPP2 interacts with the unique N-terminal domain of SK2-L that is required for synaptic localization (*Allen et al., 2011*).

## MPP2 is a synaptic MAGUK protein

To examine the anatomical localization of MPP2, the anti-MPP2 antibodies were used for immuno-histochemistry on hippocampal sections. At the light level, MPP2 was expressed throughout the hippocampus, and in area CA1, MPP2 was predominantly expressed in the dendritic arbors (*Figure 3A, B*). To determine the subcellular profile of MPP2 expression, pre- and post-embedding immuno-gold electron microscopy (iEM) was performed. Using pre-embedding techniques, sections from three animals revealed immunoparticles for MPP2 prominently labeled dendritic spines and dendritic shafts both along the plasma membrane and at intracellular sites. Post-synaptically, most immuno-particles for MPP2 were found in spines (84%; 436 particles) versus in dendrites (16%; 82 immuno-particles), with most immunoparticles in either compartment localized to the plasma membrane (84%; 365 immunoparticles in spines and 65%; 54 immunoparticles in dendrites). Next, post-embed-ding techniques were applied to determine if MPP2 is expressed in the post-synaptic density. Sec-tions from three animals revealed that immunoparticles for MPP2 prominently labeled the post-synaptic densities of dendritic spines, as well as being detected in dendritic shafts (*Figure 3C,D*). Immunoparticles for MPP2 were not detected in *stratum pyramidale*, and were also absent pre-syn-aptically. Therefore, MPP2 is a synaptic MAGUK protein.

## MPP2 is required for synaptic SK2-containing channel function

To test whether MPP2 expression is important for synaptic SK2 channel function, two short hairpin RNAs (shRNAs) targeting the 3' untranslated region (3' UTR) of *Mpp2* mRNA were co-expressed in area CA1 by in utero electroporation (e14-16) of a plasmid that also directed expression of the fluo-rescent protein, GFP. Four- to five-week-old mice were then used to prepare fresh hippocampal

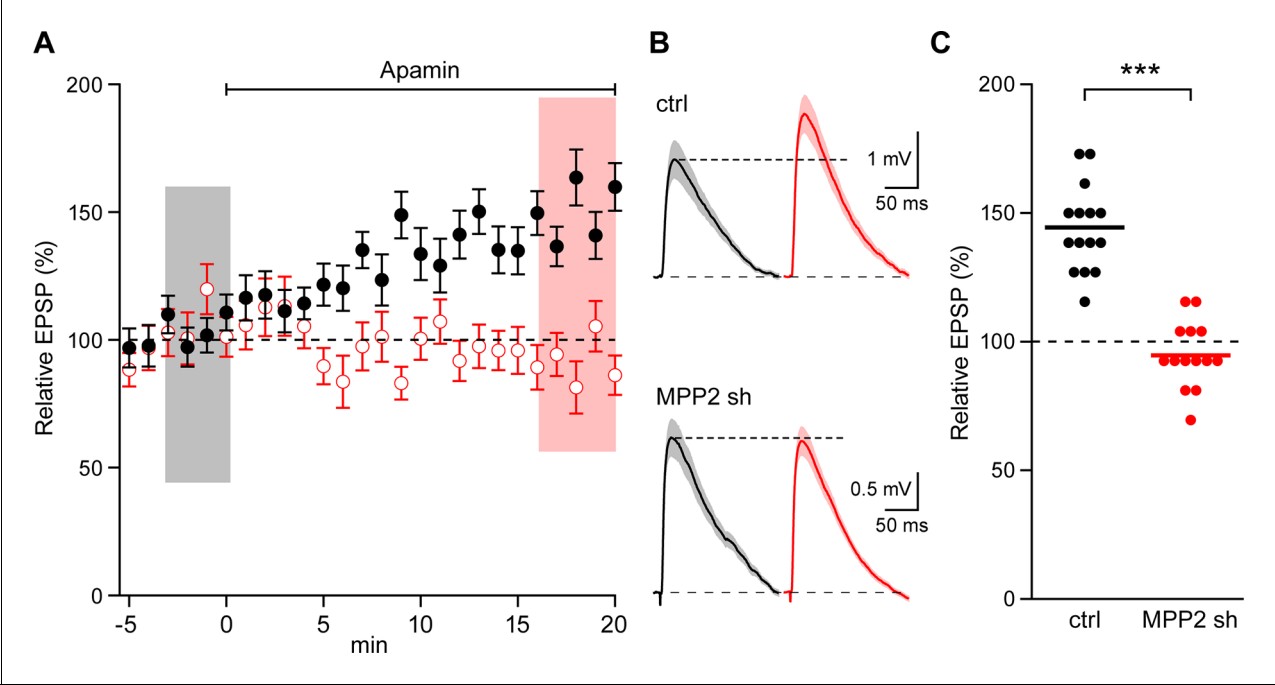

**Figure 4.** MPP2 is required for synaptic SK2-containing channel function. (**A**) Time course of the normalized EPSP amplitude (mean ± s.e.m.) for baseline in control ACSF (Ctrl) and during wash-in of apamin (100 nM) as indicated above in MPP2 sh-transfected cells (open red symbols, n = 14) and non-fluorescent control cells (black symbols, n = 15) mice. (**B**) Average of 15 EPSPs taken from indicated shaded time points in aCSF (black) and 16–20 min after application of apamin (red); shaded areas are mean ± s.e.m for control non-fluorescent cells (ctrl, upper traces) and MPP2 sh-transfected cells (MPP2 sh, bottom traces). (**C**) Scatter plot of relative ESPS peak compared to baseline from the individual slices in panel A non-fluorescent control (ctrl, black symbols) and for MPP2 sh transfected (red symbols). Horizontal bar reflects mean response.

slices. Whole-cell current clamp recordings were made from CA1 pyramidal neurons, either trans-fected, as reported by GFP expression, or non-transfected control cells. Synaptic stimulations of the Schaffer collateral axons evoked EPSPs. After establishing a stable baseline, apamin (100 nM) was applied to the slices. For control cells, apamin increased EPSPs to 144.4 ± 4.3% of baseline (n = 15; p<0.0001), consistent with previous studies (*Ngo-Anh et al., 2005*; *Lin et al., 2008*; *Giessel and Sabatini, 2011*). In contrast, apamin did not significantly affect EPSPs from transfected CA1 pyrami-dal neurons (94.6 ± 3.3%; n = 14) (*Figure 4A–C*).

To further verify the efficacy of MPP2 knock-down, pre-embedding iEM was performed on hippo-campal sections from MPP2 shRNA-transfected animals. To identify transfected CA1 pyramidal neu-rons, mouse anti-GFP antibody was detected using HRP-linked anti-mouse secondary antibody, and rabbit anti-MPP2 antibody was detected using immunogold particles coupled to anti-rabbit second-ary antibody. Immunoparticles were then counted in the same number of profiles belonging both to dendritic spines and dendritic shafts on single sections. In spines on neurons expressing GFP, immu-noparticles for MPP2 were reduced by 81% compared to spines on non-transfected control neurons (GFP-negative: 417 particles in 60 spines; GFP-positive: 101 particles in 60 spines; p<0.001). A simi-lar reduction (78%) was seen in dendritic shafts (GFP-negative: 155 particles in 25 dendrites; GFP-positive: 44 particles in 25 dendrites; p<0.001) (*Figure 5A–D*).

To demonstrate that the effects of shRNA transfection were specifically due to MPP2 knock-down, the shRNA (GFP) plasmid was co-transfected with a plasmid that directed expression of shRNA-immune MPP2 and a plasmid that expressed the red fluorophore, mApple. In doubly trans-fected neurons, apamin increased EPSPs similar to control, non-transfected neurons (148.2 ± 4.2%; n = 13; p<0.0001, compared to shRNA knock-down) (*Figure 6A,B*). Thus, the consequences of the shRNA expression are mediated by knock-down of MPP2.

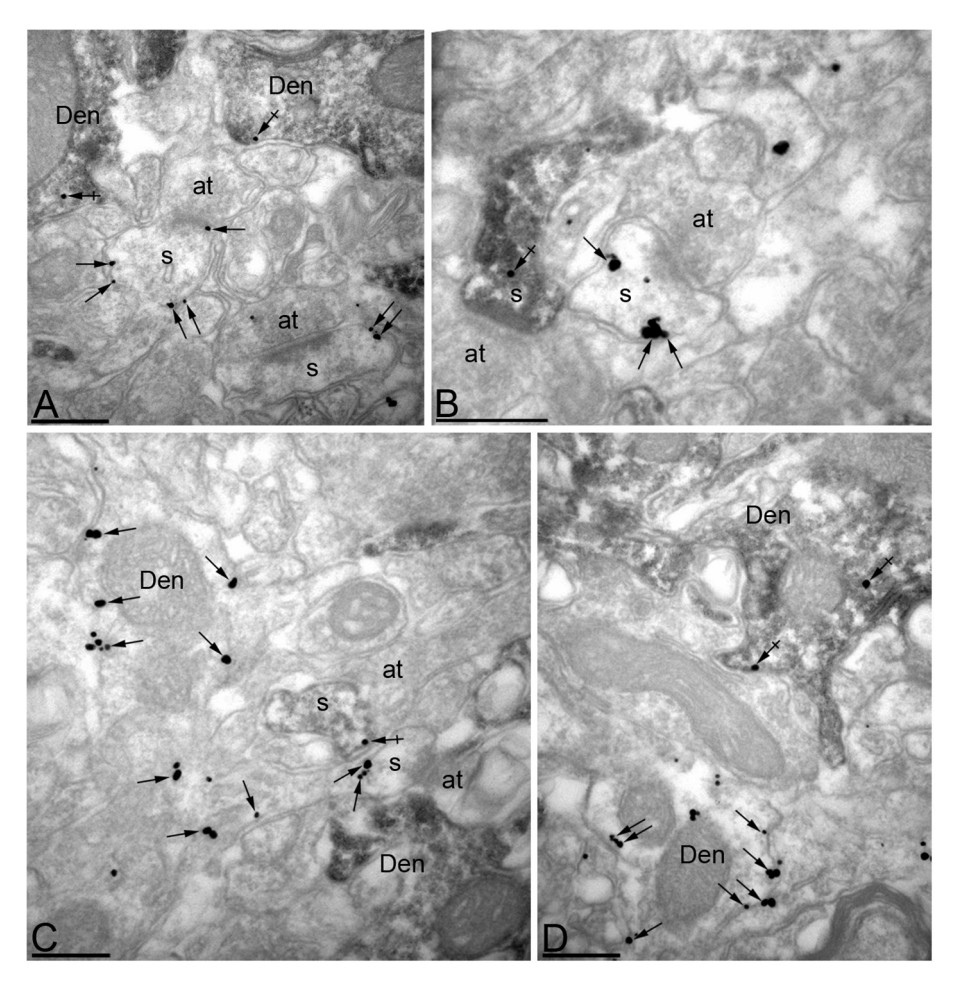

**Figure 5.** Efficient knock-down of MPP2 expression in CA1 pyramidal neurons. Immunoreactivity for MPP2 in the CA1 region of the hippocampus, as revealed using a double-labelling pre-embedding method. (**A–D**) The peroxidase reaction end product (HRP) indicating GFP immunoreactivity filled CA1 pyramidal cells, whereas immunoparticles for MPP2 were mainly located along the plasma membrane and at intracellular sites of pyramidal cells. Immunoparticles for MPP2 were distributed in both GFP-positive (crossed arrows) and GFP-negative (arrows) dendritic spines (s) and dendritic shafts (Den) of pyramidal cells. However, there was a striking reduction of immunoparticles for MPP2 in GFP-positive profiles compared to GFP-negative profiles (see text). at, axon terminal. Scale bars in **A-D**: 0.2 μm.

## MPP2 selectively affects synaptic SK2-containing channels

There are at least two functionally distinct populations of apamin-sensitive SK2-containing channels in CA1 pyramidal neurons. One population resides in the spines and is activated by synaptically evoked NMDAR-dependent $Ca^{2+}$ influx (*Ngo-Anh et al., 2005*). Another population resides in the dendrites and may be activated by somatic voltage steps that induce $Ca^{2+}$ influx through voltage-gated $Ca^{2+}$ channels (*Stocker et al., 1999*; *Gerlach et al., 2004*; *Bond et al., 2004*). To determine if MPP2 knock-down specifically affects synaptic SK2-containing channels or also affects SK2-containing channels in the dendrites transfected or control neurons were clamped at −55 mV and stepped to 20 mV; tail currents were elicited upon stepping back to −55 mV (*Figure 6A*). The apamin-sensitive component of the tail current (*Figure 7A* inset) effectively measures the SK2-containing channel component activated by somatic depolarization and showed that MPP2 knock-down did not significantly affect somatic activation of SK2-containing channels (ctrl: 41.9 ± 8.3 pA, n = 13; transfected: 53.7 ± 10.5, n = 7; P = 0.44) (*Figure 7A,B*). These results show that knocking down MPP2 expression

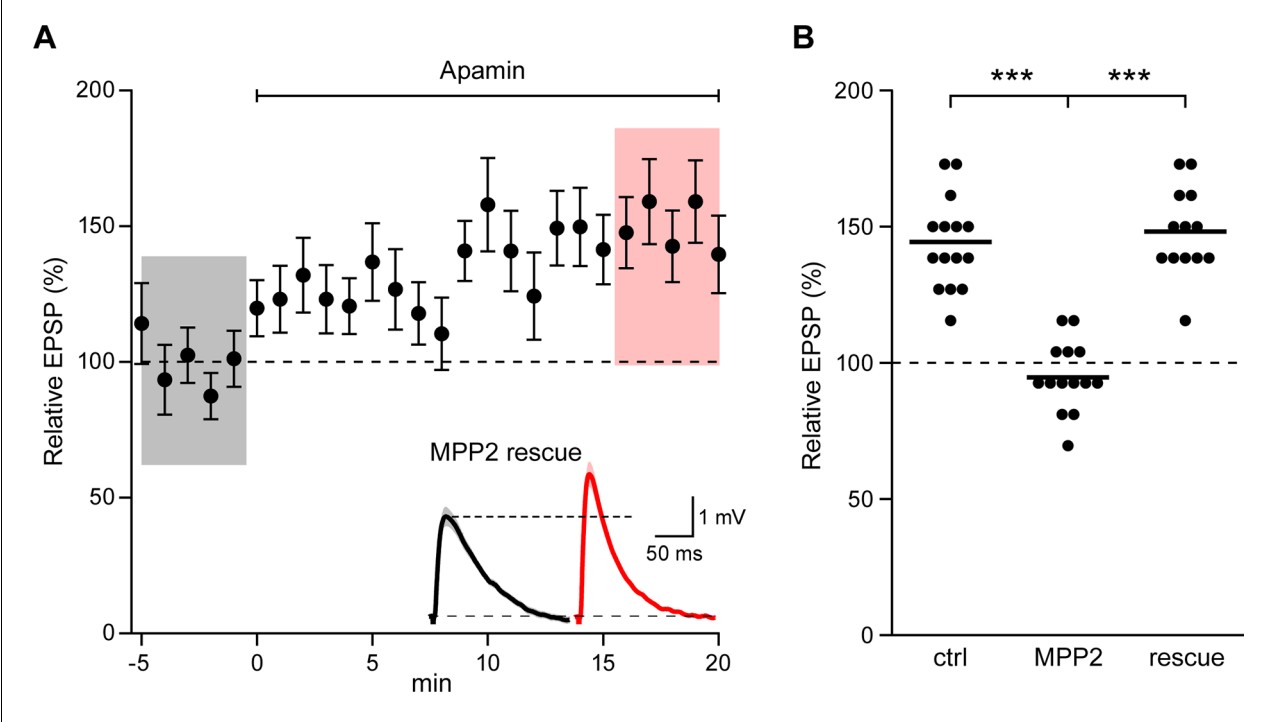

**Figure 6.** Co-expression of sh-immune MPP2 with MPP2 shRNA rescues synaptic SK2 function. (**A**) Time course of the normalized EPSP amplitude (mean ± s.e.m.) for baseline in control ACSF and during wash-in of apamin (100 nM) as indicated above in cells transfected MPP2 sh and MPP2 sh immune (n = 13). Inset: representative cell showing average of 15 EPSPs taken from indicated shaded time points in ACSF (black) and 16–20 min after application of apamin (red); shaded areas are mean ± s.e.m. (**B**). Scatter plot of relative ESPS peak compared to baseline from the individual slices for non-fluorescent control, MPP2 sh-transfected cells and MPP2 sh transfected with immune MPP2 (rescue). Horizontal bar reflects mean.

disrupts the synaptic SK2-containing channel component but does not affect the channels in the dendrites that are activated by somatic voltage pulses.

## Loss of MPP2 reduces LTP

SK channel activity modulates the induction of synaptic plasticity (*Stackman et al., 2002*) and the activity-dependent endocytosis of synaptic SK2-containing channels contributes to the expression of LTP (*Lin et al., 2008*). To determine the consequences of MPP2 knock-down and loss of synaptic SK2-containing channel activity on LTP, a theta burst pairing protocol was delivered in which Schaffer collateral stimulations were paired with back-propagating action potentials evoked by somatic current injections. In non-transfected CA1 pyramidal neurons, this induced LTP of 407.9 ± 46.3% (n = 8), while in transfected, MPP2 knock down neurons, LTP was significantly reduced (243.0 ± 14.8%; n = 13; p<0.001) (*Figure 8*). Consistent with the loss of synaptic SK2-containing channels, MPP2 knock-down reduces LTP.

## Discussion

The results presented here identify the synaptic MAGUK protein, MPP2 (p55), that is required for synaptic SK2-containing channel function. Synaptically evoked EPSPs in CA1 pyramidal neurons are increased by the SK channel blocker, apamin, but in CA1 pyramidal neurons expressing shRNAs directed against *Mpp2* mRNA apamin has no effect. MPP2 knockdown selectively affects synaptic SK2-containing channel function, as the SK2-containing channels expressed in the dendrites that are activated by somatic voltage steps are not altered. Consistent with the effects of MPP2 on synaptic SK2-containing channels, MPP2 knock-down reduces the expression of TBP-induced LTP by ~30%. This is slightly more than the component of LTP attributed to SK2 endocytosis in untransfected CA1

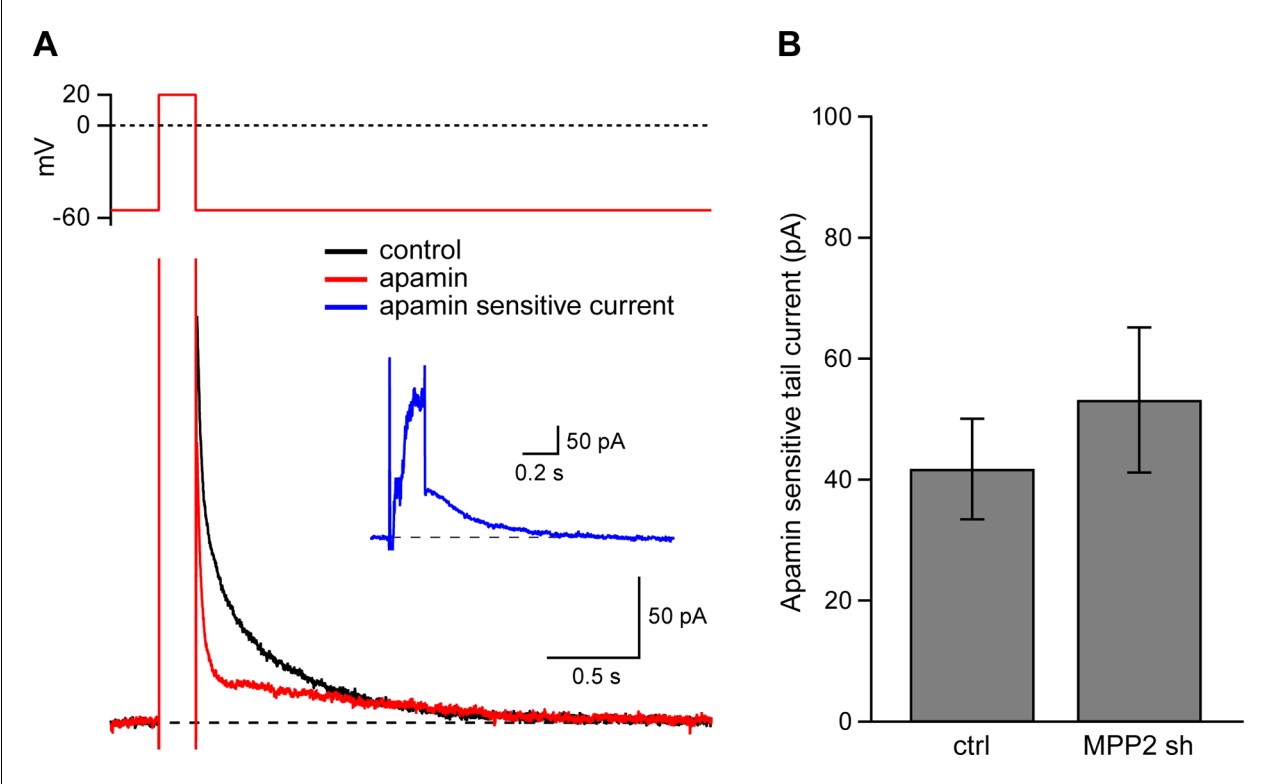

**Figure 7.** Dendritic SK channel function is not reduced by MPP2 knock-down. (**A**) Representative traces of voltage-clamp recordings of IAHP after a 200 ms depolarizing pulse to 20 mV in an MPP2 sh-transfected cell. Apamin (red trace) blocks a component of the IAHP. Inset: subtraction of the traces before and after apamin application yielded the apamin-sensitive ImAHP. (**B**) Bar graph of apamin-sensitive tail current measured at 100 ms following repolarization to −55 mV. Data presented as mean ± s.e.m. for control non-fluorescent cells (n = 13) and MPP2 sh-transfected cells (n = 7).

pyramidal neurons, ~17% (*Lin et al., 2008*). This might reflect effects of MPP2 knock-down on other interaction partners (see below).

Previous results showed that synaptic SK2-containing channels are heteromeric assemblies that contain two isoforms of SK2, SK2-S and SK2-L. Compared to SK2-S, SK2-L has an additional 207 amino acids in the intracellular N-terminal domain and SK2-S is otherwise entirely contained in SK2-L (*Strassmaier et al., 2005*). In a transgenic mouse selectively lacking SK2-L, the SK2-S channels are expressed in the dendrites and even in the plasma membrane of dendritic spines, but they are specifically excluded from the PSD, and apamin fails to boost synaptically evoked EPSPs. Re-expressing SK2-L reinstates synaptic function as measured by apamin sensitivity of EPSPs (*Allen et al., 2011*). These results implicated the unique N-terminal domain of SK2-L in directing SK2-containing channel synaptic localization and function, and suggested that the N-terminal domain of SK2-L might interact with a partner protein to engender PSD localization. Indeed, MPP2 binds to the unique N-terminal domain and knocking down MPP2 expression phenocopied the effect of SK2-L deletion on synaptic responses. MAGUK proteins recruit and stabilize AMPA and NMDA receptors in the PSD. Our results suggest that MPP2 similarly serves to stabilize SK2-containing channels in the PSD.

MPP2 is a member of the p55 Stardust subfamily of MAGUK scaffold proteins, named after the founding member MPP1, the major palmitoylated protein in erythrocytes (*Alloisio et al., 1993*). Similar to other MAGUK scaffold proteins, MPP2 is modular, consisting of two L27 domains that may mediate homo- or heterophilic interactions, a single PDZ domain followed by SH3-HOOK-GK domains. Biochemical studies showed that MPP2 is enriched in the PSD fractions of rat brain, and pull-down assays to test for interactions suggested MPP2 may interact with itself as well as a number of other synaptic proteins, among them are PSD-95, SAP97, GKAP, CASK, GRIP, neuroligin, and

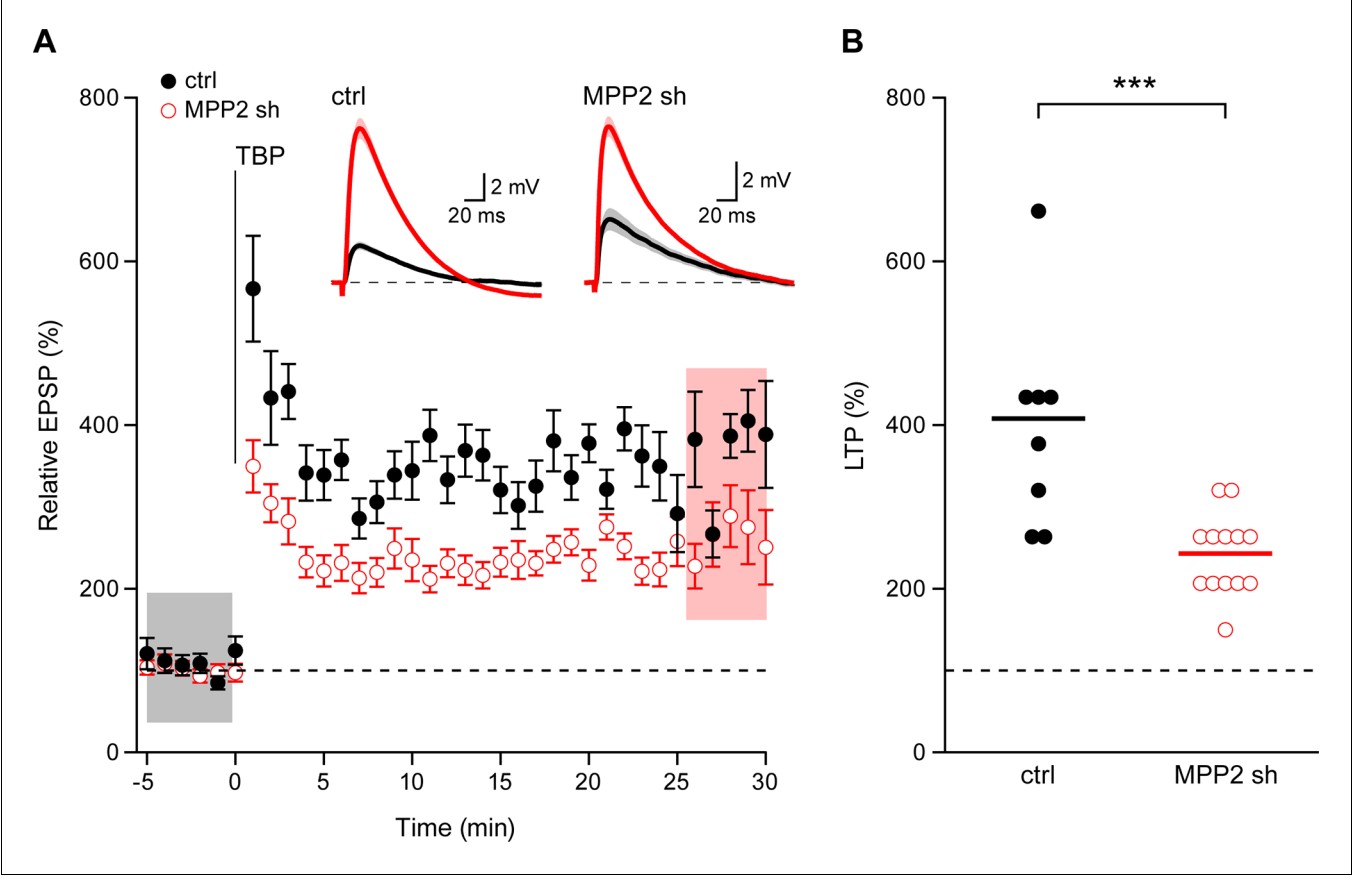

**Figure 8.** Loss of MPP2 reduces LTP. (**A**) Time course of the normalized EPSP amplitude (mean ± s.e.m.) from control non-fluorescent cells (ctrl, closed black symbols, n = 8) and MPP2 sh-transfected cells (MPP2, open red symbols, n = 13). The TBP protocol was delivered at time 0. Inset: representative cell showing average of 15 EPSPs taken from indicated shaded time points in ACSF (black) and 25–30 min after the induction of LTP (red); shaded areas are mean ± s.e.m. (**B**). Scatter plot of relative ESPS peak compared to baseline from the individual slices for non-fluorescent control (ctrl) and MPP2 sh-transfected cells. Horizontal bar reflects mean response.

CaMKII. The SH3-HOOK-GK domain of MPP2 was implicated in mediating these interactions (*Jing-Ping et al., 2005*), similar to the interaction between MPP2 and the N-terminal domain of SK2-L.

The results presented here, using an unbiased approach also identified SK3 as immunopurifying with MPP2. SK2 and SK3 can form heteromeric channels in brain (*Strassmaier et al., 2005*), and SK3 is expressed in CA1 pyramidal neurons (*Ballesteros-Merino et al., 2014*). Different from SK2 that has two N-terminal isoforms, there is only one SK3 N-terminal isoform and it is similar to the extended N-terminal domain of SK2-L, harboring several islands of homology that might mediate interactions with MPP2.

The proteomics analyses of MPP2 also identified three additional scaffold proteins, the MAGUK protein, DLG1 (SAP97), as well as Lin7A and Lin7C as MPP2-interacting proteins. In epithelial cells, MPP7, a closely related member of the p55 Stardust family, dimerizes with Lin7 proteins, an interaction mediated by the C-terminal L27 domain, and the dimeric complex then associates with DLG1 via the N-terminal L27 domain that is insufficient to mediate DLG1 interactions in the absence of bound Lin7 (*Bohl et al., 2007*). DLG1 also binds the C-terminal PDZ ligand on the GluA1 subunit of AMPA receptors (*Leonard et al., 1998*). Indeed, upon the induction of LTP at Schaffer collateral to CA1 synapses, additional GluA1-containing AMPA receptors undergo exocytosis at a perisynaptic site followed by translocation into the post-synaptic membrane, increasing the AMPA component of EPSPs (*Yang et al., 2008*). This exocytosis is dependent on the PDZ ligand at the C-terminus of the GluA1 subunit (*Lin et al., 2010*; *Yang et al., 2008*; *Shi et al., 2001*), and exocytosis of GluA1-containing AMPA receptors is prerequisite to SK2-containing channel endocytosis; specifically blocking

GluA1-containing AMPA receptor exocytosis prevents the rapid, subsequent endocytosis of SK2-containing channels (*Lin et al., 2010*). Moreover, immunopurification of AMPA receptors from whole brain identified MPP2 as a protein that co-purified with AMPA receptors (*Schwenk et al., 2012*). Synaptic SK2-containing channels reside in very close proximity to synaptic NMDA receptors within the PSD, providing a molecular microdomain that facilitates their functional coupling (*Ngo-Anh et al., 2005*; *Lin et al., 2008*). PSD-95 interacts with NMDARs (*Kornau et al., 1995*), is crucial for the proper synaptic localization of ionotropic glutamate receptors (*Schnell et al., 2002*; *Elias et al., 2008*), and interacts with SAP97 (*Cai et al., 2006*). It will be interesting to determine whether MPP2 additionally interacts, directly or indirectly with PSD-95 to maintain the spatial synaptic relationship between SK2-containing channels and NMDA receptors. MPP2 contains multiple different protein-protein interaction domains that may bind not only receptors and channels but, additionally, signaling molecules as well as connections to the cytoskeleton. These observations raise the possibility that there is a dynamic protein lattice encompassing SK2-containing channels, AMPARs and NMDARs, and regulatory proteins that is woven together by molecular interactions between scaffold proteins to precisely tune synaptic responses during basal neurotransmission and plasticity.

## Materials and methods

### Animal care

All procedures involving animals were performed in accordance with the guidelines of Oregon Health and Science University (Portland, OR), animal care protocol number: IP00000191; University of Freiburg (Freiburg, Germany), Regierungspräsidium Freiburg, AZ: 35–9185/G-12/47; and University of Castilla-La Mancha (Albacete, Spain).

### Molecular biology

MPP2 shRNAs were designed using an online algorithm (http://sirna.wi.mit.edu). Two sequences targeting the 3'UTR of mouse *Mpp2* mRNA (NM_016695; 1811–1833; 3620–3642) were synthesized for shRNA expression and cloned into a vector that drove their expression from the U6 promoter. This same plasmid also directed GFP expression from the ubiquitin promoter. For rescuing MPP2 knockdown, the MPP2 shRNA plasmids (GFP) were co-transfected with a plasmid expressing the MPP2 coding sequence, cloned between the CAG promoter and the 3' UTR from bovine growth hormone. This plasmid was co-transfected with a plasmid directing mApple expression from the ubiquitin promoter.

### Antibodies

To generate antibodies to MPP2, rabbits were immunized with synthetic peptides representing amino acids 116–145 and 480–507 of mouse MPP2 (NP_057904). To test the antibodies using Western blots, mouse whole brains were homogenized in ice-cold homogenization buffer (0.32 M sucrose, 1 mM ethylenediaminetetraacetic acid, 1 mM ethyleneglycoltetraacetic acid, 10 mM Tris-HCl, pH 7.2, 0.4 mM phenylmethylsulfonyl fluoride). The resultant homogenate was subjected to centrifugation at 1000 $g$ for 10 min to remove nuclei and debris. The protein concentration was determined with Lowry's method. Human embryonic kidney 293T (HEK293T) cells were transfected with pFLAG-CMV vector (Clontech, Palo Alto, CA) encoding mouse FLAG-MPP2 in a 10 cm dish. Cells were harvested in 0.5 ml of PBS. Homogenates and suspended cells were mixed with an equal volume of 2 x sodium dodecyl sulfate (SDS) sampling buffer (63 mM Tris-HCl, pH 6.8, 4% SDS, 20% glycerol, 0.002% bromophenol blue), and denatured with 50 mM ( ± )-dithiothreitol at 55°C for 30 min. Proteins (100 µg of brain homogenates and 0.5 µl of cell lysates) were separated using 10% SDS-polyacrylamide gel electrophoresis, and electroblotted onto an Immobilon-P Transfer Membrane (Millipore, Billerica, MA). After blocking with 5% skimmed milk for 30 min, blotted membranes were incubated with primary antibodies (1 µg/ml) for 1 hr, then with peroxidase-conjugated secondary antibodies for 1 hr (Jackson ImmunoResearch, West Grove, PA; 1:10,000). Tris-buffered saline (10 mM Tris-HCl, pH 7.5, 150 mM NaCl) containing 0.1% Tween-20 was used as the dilution and washing buffer. Immunoreactions were visualized with the ECL chemiluminescence detection system, and captured using an ImageQuant LAS 500 (GE Healthcare, Buckinghamshire, UK). For specificity

control, anti-MPP2a and anti-MPP2b (1 µg/ml) were mixed with 20 µg/ml of GST fusion proteins of the respective immunizing antigen.

Anti-SK2 antibodies have been previously characterized (*Lin et al., 2008*; *Allen et al., 2011*; *Ballesteros-Merino et al., 2012*). Mouse monoclonal anti-Myc (05–419), mouse monoclonal anti-GST (05–311), rabbit polyclonal anti-GluA1 (AB1504), and mouse monoclonal anti-GluA1 (MAB2263) antibodies were from EMD Millipore (Billerica, MA). Anti-6XHis mouse monoclonal antibody (37–2900) and anti-PSD-95 mouse monoclonal antibody (MA1-045) were from Thermo Scientific (Waltham, MA). Anti-GFP (ab92) mouse monoclonal antibody was from Abcam (Cambridge, MA). Mouse monoclonal C8 antibody was obtained as supernatant from Chessie 8 cells, LN 10300SP (National Cell Culture Center, Biovest, Minneapolis, MN). Secondary antibodies were from Santa Cruz Biotechnology (Dallas, TX), unless otherwise noted.

## Proteomic analysis

Proteomic analysis of MPP2 from rat brain was performed as described previously (*Schwenk et al., 2012*; *Schwenk et al., 2010*; *Bildl et al., 2012*). Briefly, affinity-purifications (APs) with anti-MPP2a, anti-MPP2b, and pre-immunization immunoglobulins (IgG) were performed on rat brain membrane fractions solubilized with CL-91 (Logopharm, March, Germany) and whole eluates were subjected to high-resolution mass spectrometry. Proteins retained in APs and identified by mass spectrometry were quantified by integration of peptide m/z signal intensities over time (peak volumes, PVs) that were extracted from FT full scans using MaxQuant (v.1.4.1.2; http://www.maxquant.org; with integrated off-line mass calibration). Relative abundance of proteins in anti-MPP2 samples versus control (abundance ratio or rPV) was determined by the TopCorr method (*Bildl et al., 2012*) as the median of either (i) six individual peptide PV ratios of the best correlating protein-specific peptides (as determined by Pearson correlation of their abundance values; TopCorr6), or (ii) the 50% best of all individual peptide PV ratios (Top50). The linear dynamic range of the TopCorr method is about 4 orders of magnitude (detailed in [*Bildl et al., 2012*]). The data used for rPV determination (including peptide sequences, PV and rPV values, as well as the respective medians and selections methods) are summarized in *Supplementary file 1*. The coverage of the primary sequences of all proteins shown in *Figure 1C* is presented in *Figure 1—figure supplement 1* and *Figure 1—figure supplement 2*.

## Co-immunoprecipitation

HEK293 cells were transfected using Lipofectamine 2000 (Thermo Scientific). After 48 hr cells were washed, collected with PBS buffer, and pelleted. Cells were then solubilized with lysis buffer (20 mM HEPES, pH 7.5, 150 mM NaCl, 10% glycerol, 2 mM EDTA, 1 mM PMSF, and protease inhibitors (Hoffman La Roche, Basel, Switzerland) containing 1% β-D-dodecyl maltoside (Sigma-Aldrich, St. Louis, MO) for 30 min at RT. The lysate was centrifuged at 14,000 rpm for 20 min at 4°C. The supernatant was incubated with the indicated primary antibody for 30 min then 20 µl of protein A/G agarose beads (Thermo Scientific) were added for incubation overnight at 4°C with rotation. The beads were washed three times in washing buffer (mM) (20 HEPES, pH 7.5, 150 NaCl, 10 KCl, 2 EDTA, 10% glycerol). Proteins were eluted with 40 µl of 2X SDS sample buffer at 37°C. Bound and eluted proteins were subsequently separated by SDS-PAGE and transferred to PVDF membrane (BioRad, Hercules, CA). After blocking with 5% skimmed milk for 1 hr, the membrane was probed with the indicated antibodies over night at 4°C. HRP-conjugated secondary antibodies were applied for 30 min at RT. Blots were detected with SuperSignal ECL (Thermo Scientific) and developed with GeneMate Blue film (BioExpress, Kaysville, UT).

## GST pull-downs

The N-terminal domain of SK2-L (NP_001299834; 67–273) was cloned into pET33b for His tagged expression. Plasmids for expression of His-tagged rat PSD-93 (P85-T280) and GST-Kv1.4-ct (H570-V665) were gifts from Dr. Paul Slesinger (*Lunn et al., 2007*). For GST-fusion proteins, the SH3-HOOK-GK domains of mouse MPP2 (216–552; NM_016695.3), SAP97 (577–927; NM_007862.2), and CaCNB4 (45–371; NM146123.2) were cloned into pGex4T3. Expression levels of the GST-baits were determined by Coomassie staining and Western blotting with anti-GST antibody (*Figure 2—figure supplement 2*), or for His-prey, with anti-His antibody (*Figure 2*). GST pull-downs were performed as previously described (*Allen et al., 2007*), with minor modifications. Glutathione agarose beads

(Sigma-Aldrich) were resuspended and 50 µl of slurry was used per reaction. Beads were washed one time with pull-down wash buffer (PWB; 20 mM HEPES, pH 7.8, 10% glycerol, 100 mM KCl, 0.1 mM EDTA, 0.1 mM dithiothreitol [DTT], 0.1% Igepal CA-630 [Sigma-Aldrich]) and then incubated at 4°C for 2 hr with bacterial lysate containing the GST-fusion protein. Bead–protein complexes were washed one time with PWB followed by a 5 min wash at 4°C with PWB plus 0.1% BSA, and two 1 min washes with PWB. These 'baits' were incubated at 4°C overnight with bacterial cell lysate from cultures expressing the His-tagged 'prey', washed four times with PWB, added to SDS sample buffer, heated at 95°C for 5 min and resolved by SDS-PAGE. Following transfer the Western blot was probed with anti-His antibody.

## Immunoelectron microscopy

Immunohistochemical reactions at the electron microscopic level were carried out using immunogold methods as described previously (Luján et al., 1996).

### Pre-embedding immunohistochemistry

Single- and double-labelling methods were used. Briefly, in single-labelling experiments, free-floating sections were incubated in 10% NGS diluted in TBS. Sections were then incubated in anti-MPP2 antibodies (1–2 µg/ml diluted in TBS containing 1% NGS), followed by incubation in goat anti-rabbit IgG coupled to 1.4 nm gold (Nanoprobes, Yaphank, NY). Sections were postfixed in 1% glutaraldehyde and washed in double-distilled water, followed by silver enhancement of the gold particles with an HQ Silver kit (Nanoprobes). In co-labelling experiments, GFP immunoreactivity was visualized by the immunoperoxidase reaction, and MPP2 immunoreactivity was revealed with the silver-intensified immunogold reaction. Sections were then treated with osmium tetraoxide (1% in 0.1 M PB), block-stained with uranyl acetate, dehydrated in graded series of ethanol and flat-embedded on glass slides in Durcupan (Sigma-Aldrich) resin. Regions of interest were cut at 70–90 nm on an ultramicrotome (Reichert Ultracut E, Leica Biosystems, Barcelona, Spain) and collected on 200-mesh copper grids. Staining was performed on drops of 1% aqueous uranyl acetate followed by Reynolds's lead citrate.

### Post-embedding immunohistochemistry

Briefly, ultrathin 80-nm-thick sections from Lowicryl-embedded blocks of the hippocampus were picked up on coated nickel grids and incubated on drops of a blocking solution consisting of 2% human serum albumin in 0.05 M TBS and 0.03% Triton X-100. The grids were incubated with MPP2 antibodies (10 µg/ml in 0.05 M TBS and 0.03% Triton X-100 with 2% human serum albumin) at 28°C overnight. The grids were incubated on drops of goat anti-rabbit IgG conjugated to 10 nm colloidal gold particles (Nanoprobes) in 2% human serum albumin and 0.5% polyethylene glycol in 0.05 M TBS and 0.03% Triton X-100. The grids were then washed in TBS and counterstained for electron microscopy with saturated aqueous uranyl acetate followed by lead citrate.

Ultrastructural analyses were performed in a Jeol-1010 electron microscope (Synaptic Structure Laboratory, School of Medicine, University of Castilla-La Mancha, Albacete, Spain). Electron photomicrographs were captured with an ORIUS SC600B CCD camera (Gatan, Munich, Germany). Digitized electron images were modified for color, brightness, and contrast with Adobe Photoshop, version 7.0. Labelled structures were classified based on unambiguous morphological information in each section. Axon terminals were identified by the presence of synapses and small round and/or large granular vesicles. Synapses were identified as parallel membranes separated by widened clefts that were associated with membrane specializations. Synapses displaying a prominent density on the post-synaptic side were characterized as asymmetrical. Dendritic spines were identified as small protrusions exhibiting membrane continuity with the dendritic shaft. Significance of immunogold particle labelling in GFP-positive versus GFP-negative spines and dendrites was assessed by ANOVA.

## In utero electroporation

Timed-pregnant mice were anesthetized with isofluorane, their abdominal cavity cut open, and the uterine horns/sac exposed. Approximately 2 µl of DNA solution (~2 mg/ml) was injected into the lateral ventricle of e14-e16 embryos, using a glass pipet pulled from thin walled capillary glass (TW150F-4, World Precision Instruments, Sarasota, FL) and a Picospritzer III microinjection system

(Parker Hannifin, Hollix, NH). The head of each embryo within its uterine sac was positioned between tweezer-type electrodes (CUY650P10; Sonidel, Dublin, Ireland), and 5 square electric pulses (35 V; 50 ms; 1-s intervals) were passed using an electroporator (CUY21; Sonidel). After electroporation, the wall and skin of the abdominal cavity of the pregnant mouse was sutured-closed, and embryos were allowed to develop normally.

## Slice preparation

Hippocampal slices were prepared from 4- to 5-week-old C57BL/6J mice. Animals were anesthetized with isoflurane and decapitated. The cerebral hemispheres were quickly removed and placed into cold artificial CSF (ACSF) equilibrated with carbogen (95% $O_2$/5% $CO_2$). Hippocampi and cortex were removed, placed onto an agar block, and transferred into a slicing chamber. Transverse hippocampal slices (300–350 µm) were cut with Leica VT1200s (Leica Biosystems, Buffalo Grove, IL) and transferred into a holding chamber containing regular ACSF. Slices were incubated at 34°C for 30 min and then at room temperature for $\geq 1$ hr before recordings were performed. The slicing solution consisted of sucrose-ACSF (in mM): 70 sucrose, 80 NaCl, 2.5 KCl, 21.4 $NaHCO_3$, 1.25 $NaH_2PO_4$, 0.5 $CaCl_2$, 7 $MgCl_2$, 1.3 ascorbic acid, 20 glucose and regular ACSF consisted of (in mM): 125 NaCl, 2.5 KCl, 21.5 $NaHCO_3$, 1.25 $NaH_2PO_4$, 2.0 $CaCl_2$, 1.0 $MgCl_2$, 15 glucose, equilibrated with carbogen.

## Electrophysiology

CA1 pyramidal cells were visualized with IR/DIC optics (Olympus BX51W1; Olympus Scientific Solutions, Waltham, MA) and a CCD camera. Whole-cell, patch-clamp recordings were obtained from CA1 pyramidal cells using a Multiclamp 700B (Molecular Devices, Sunnyvale, CA), digitized using an ITC-18 analog-to-digital converter, and transferred to a computer using Patchmaster software (Heka Instruments, Bellmore, NY). Patch pipettes (open pipette resistance, 2–4 MΩ) were filled with (in mM) 133 K-gluconate, 4 KCl, 4 NaCl, 1 $MgCl_2$, 10 4-(2-hydroxyethyl)-1-piperazineethanesulfonic acid (HEPES), 4 MgATP, 0.3 $Na_3$ guanosine triphosphate ($Na_3$GTP), and 10 $K_2$-phosphocreatine (pH 7.3). Electrophysiological records were filtered at 3.3 kHz and sampled at 10 kHz. Series resistance was electronically compensated to greater than 70%. A bias current was applied to maintain the membrane potential in current clamp at −65 mV. The input and residual series resistance in current clamp was determined from a 20 pA hyperpolarizing pulse applied at the end of each sweep. The input resistance in voltage clamp was determined from a 5 mV hyperpolarizing pulse applied at the beginning of each sweep. All recordings were from cells with a resting membrane potential less than −60 mV and a stable input resistance. All electrophysiological recordings were performed at 22–24°C, and data were not corrected for a junction potential of ~15 mV of the internal solution with respect to the bath ACSF.

## Synaptic stimulation

EPSPs were recorded in whole-cell mode. Capillary glass pipettes (tip diameter, ~5 µm) filled with ACSF and connected to Digitimer constant current stimulus isolation unit (AutoMate Scientific, Berkeley, CA) were used to stimulate pre-synaptic axons in *stratum radiatum* as described in Results. SR95531 (2–10 µM) and CGP55845 (1 µM) were present to reduce $GABA_A$ and $GABA_B$ contributions, respectively. To prevent epileptic discharges in the presence of GABAergic blockers, the CA3 region was microdissected out of slices used for EPSP recordings. LTP was induced by a theta burst pairing protocol, as previously described (*Lin et al., 2008*).

## Data analysis

Data were analyzed using IGOR (WaveMetrics, Lake Oswego, OR). Data are expressed as mean ± SEM. Paired two-sample *t*-tests were used to determine significance of data in the same pathway, and non-parametric Wilcoxon Mann-Whitney two-sample rank test was used to determine significance between groups of data; $p < 0.05$ was considered significant.

## Chemicals and solutions

D-AP5, CNQX, CGP55845, and SR95531 were obtained from Tocris Bioscience (Ellisville, MO). All other chemicals were obtained from Sigma-Aldrich unless specified. All perfusing solutions were modified from regular ACSF unless otherwise noted.

## Acknowledgements

We thank Dr. Paul Slesinger for Kv1.4 and PSD-93. We also thank Ms. Lori Vaskalis for expert graphics design. This work was supported by grants from the NIH to JM and JPA (MH093599 and NS038880) and to RL from the Spanish Ministry of Education and Science (BFU-2012-38348), European Union (HBP - Project Ref. 604102) and Junta de Comunidades de Castilla-La Mancha (PPII-2014-005-P).

## Additional information

### Funding

| Funder | Grant reference number | Author |
|---|---|---|
| Spanish Ministry of Education and Science | BFU-2012-38348 | Rafael Luján |
| European Union | HBP - Project Ref. 604102 | Rafael Luján |
| Junta de Comunicades de Castilla-La Mancha | PPII-2014-005-P | Rafael Luján |
| National Institutes of Health | MH093599 | James Maylie John P Adelman |
| National Institutes of Health | NS038880 | James Maylie John P Adelman |

The funders had no role in study design, data collection and interpretation, or the decision to submit the work for publication.

### Author contributions

GK, RL, Conception and design, Acquisition of data, Analysis and interpretation of data, Drafting or revising the article; JS, MHK, CA, MW, Acquisition of data, Analysis and interpretation of data; BF, Conception and design, Acquisition of data, Analysis and interpretation of data; JM, JPA, Conception and design, Analysis and interpretation of data, Drafting or revising the article

### Author ORCIDs

John P Adelman, http://orcid.org/0000-0002-1135-1549

### Ethics

Animal experimentation: All procedures involving animals were performed in accordance with the guidelines of Oregon Health and Science University (Portland, OR), animal care protocol number: IP00000191, University of Freiburg (Freiburg, Germany) Regierungspräsidium Freiburg, AZ: 35-9185/G-12/47, and University of Castilla-La Mancha (Albacete, Spain).

## Additional files

### Supplementary files

• Supplementary file 1. Relative protein abundance in a sample versus control.

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
