## [Decision Letter]

Thank you for submitting your work entitled "Membrane palmitoylated protein 2 is a synaptic scaffold protein required for synaptic SK2-containing channel function" for consideration by *eLife*. Your article has been reviewed by three peer reviewers, one of whom is a member of our Board of Reviewing Editors, and the evaluation has been overseen by Richard Aldrich as the Senior Editor.

The following individuals involved in review of your submission have agreed to reveal their identity: Leonard Kaczmarek and John Scott (peer reviewers).

The reviewers have discussed the reviews with one another and the Reviewing Editor has drafted this decision to help you prepare a revised submission.

Summary:

This is an interesting, generally well-written and straightforward article that defines a role for the synaptic scaffold protein membrane palmitoylated protein 2 (MPP2) in the modulation of synaptic SK2 channel action. The authors delineate the domains of interaction between MPP2 and the cytosolic domains of SK channels. Using an appropriate combination of proteomic, immunocytochemistry and electrophysiological techniques they show that manipulation of MPP2 affects SK2 channel modulation and synaptic long-term potentiation.

Essential revisions:

The data are not yet conclusive because there are five points that need to be more firmly established.

1) The initial finding that led to the characterization of MPP2 as an interacting partner for SK2 was a mass spec-based analysis of SK2-containing channels immunopurified from rat brain. Other than the statement in the first sentence of the Results that this finding was the key to the all of the other experiments in the entire study, nothing is reported about this experiment.

A).The authors should provide data files of the peptide peak areas that were measured by mass spectrometry and used to calculate ratios shown in Figure 1. How were those peptides chosen? How was the dynamic range established?

B) The information in the Methods section is not adequate to establish the nature or specificity of the antibodies generated against MPP2. The authors need to state what antigen they used to generate the antibodies. They need to show a Western blot made with an entire lane of fractionated brain homogenate; the lane should show the molecular weights of all bands recognized by the antibody (including MPP2), and a second lane should show that the MPP2 band is not labeled when the antibody is preincubated with the antigen. The data in Figure 1 do not establish the specificity of the antibodies against MPP2. The authors have suggested that this data indicates that SAP-97, Lin7A and/or Lin7C co-assemble with MPP2 in rodent brain. However, the data are equally compatible with the conclusion that the two antibodies cross-react with SAP-97, Lin7A and/or Lin7C. Without this demonstration of specificity, the immunocytochemistry experiments carried out in Figure 2 and Figure 4 are also not conclusive.

2) The data in Figure 1 appear to establish that: 1. the channel protein SK2 can interact with MPP2 when the two proteins are expressed in HEK cells, and 2. SK2 does not interact with PSD-95 when the two proteins are co-expressed in HEK cells. However, the data would be more convincing if the authors indicated the molecular weight of the immunoprecipitated MPP2 band, and showed an entire lane, identifying any other cross-reacting bands. In the second paragraph of the subsection “MPP2 interacts with SK2 “the authors indicate a control as "data not shown." These are important controls and the data should be presented.

3) The data in Figure 1 presented problems for two reviewers:

A) The data in Figure 1 shows that the SK2L isoform does not interact with Shank, SAP97, or CACNB4 when co-expressed; however, this data does not exclude interaction of SK2-S with SAP-97. In the subsection “MPP2 interacts with SK2”, referring to Figure 1, the authors state that "…several other MAGUK proteins (Shank1, SAP97, CaCNB4) were prepared… and used as baits for the His-tagged N-terminal domain of SK2-L." Shank1 and CaCNB4 are not MAGUK proteins and do not have an SH3-HOOK-GK domain. The experiment in Figure 1 shows that the SH3-HOOK-GK domain of SAP-97 doesn't pull down SK2-L. The significance of its lack of interaction with Shank1 and CaCNB4 is less clear.

B) A major part of the text that refers to Figure 1 (subsection “MPP2 interacts with SK2”, second paragraph) describes experiments showing that the SH3-HOOK-GK domain of MPP2 is the domain that is required for interaction with SK2-L. While Figure 1 may perhaps represent the data for this, there is no reference to Figure 1 in the text. In addition, the legend for Figure 1 makes no mention of SH3-HOOK-GK domains (but does refer to GST-pulldowns) and gives the impression that the full length proteins are being used here. The Methods do not give any information that allows one to identify what sequence is being used as bait in these experiments. Finally, the text erroneously refers to Figure 1 as showing that Kv1.4 pulls down PSD-93. In summary, either an important part of the text is missing or these experiments have to be described more clearly (or both).

4) The immunohistochemistry and EM analysis that represent the core of MPP2 analysis are well done but need to be accompanied by intermediate resolution staining that shows the subcellular location of MPP2 in synapses of cultured neurons and whether or not the protein exhibits a distinct pattern of neuronal distribution from the marker protein PSD-95.

5) The data in Figure 4 is not a convincing demonstration of specific knockdown of MPP2. The authors should demonstrate the specificity of knockdown in cultured neurons where more quantitative measures are possible and where the possibility of off-target effects can be assessed. For example, is the expression of SAP97 altered?

---

## [Author Response]

Essential revisions:

*The data are not yet conclusive because there are five points that need to be more firmly established.*

*1) The initial finding that led to the characterization of MPP2 as an interacting partner for SK2 was a mass spec-based analysis of SK2-containing channels immunopurified from rat brain. Other than the statement in the first sentence of the Results that this finding was the key to the all of the other experiments in the entire study, nothing is reported about this experiment.*

*A) The authors should provide data files of the peptide peak areas that were measured by mass spectrometry and used to calculate ratios shown in*
Figure 1*. How were those peptides chosen? How was the dynamic range established?*

Proteins retained in all anti-MPP2 APs and identified by mass spectrometry were quantified by integration of peptide m/z signal intensities over time (peak volumes, PVs). PVs were extracted from FT full scans using MaxQuant (v.1.4.1.2; http://www.maxquant.org; with integrated off-line mass calibration).

Relative protein abundance in a sample versus control (rPV, [Supplementary-material SD1-data]) was determined by the TopCorr method (Bildl et al., MCP, 2012) as the median of either (i) six individual peptide PV ratios of the best correlating protein-specific peptides (as determined by Pearson correlation of their abundance values; TopCorr6) or (ii) the 50% best of all individual peptide PV ratios (Top50).

The linear dynamic range of the TopCorr method is about 4 orders of magnitude (detailed in Bildl et al., MCP, 2012).

The data used for rPV determination, including peptide sequences, PV and rPV values, as well as the respective medians and selections methods, were summarized and added to the revised manuscript as new [Supplementary-material SD1-data]. In addition, we have added as Figure 1—figure supplement 1 and Figure 1—figure supplement 2, the coverage of the primary sequences of all proteins shown in, now, Figure 1.

These details have been added to the Methods section.

*B) The information in the Methods section is not adequate to establish the nature or specificity of the antibodies generated against MPP2. The authors need to state what antigen they used to generate the antibodies.*

Done, as requested.

*They need to show a Western blot made with an entire lane of fractionated brain homogenate; the lane should show the molecular weights of all bands recognized by the antibody (including MPP2), and a second lane should show that the MPP2 band is not labeled when the antibody is preincubated with the antigen.*

This is now part of revised Figure 1.

*The data in Figure 1 do not establish the specificity of the antibodies against MPP2. The authors have suggested that this data indicates that SAP-97, Lin7A and/or Lin7C co-assemble with MPP2 in rodent brain. However, the data are equally compatible with the conclusion that the two antibodies cross-react with SAP-97, Lin7A and/or Lin7C. Without this demonstration of specificity, the immunocytochemistry experiments carried out in Figure 2 and Figure 4 are also not conclusive.*

The new Western blots (revised Figure 1) show the antibodies have high specificity for MPP2. In addition we cloned and expressed, separately, C8-tagged SAP97, Lin7A and Lin7C in HEK293 cells and performed Western blots using the MPP2 antibodies; the expressed proteins were not detected. Probing with anti-C8 antibody confirmed the proteins were expressed (revised Figure 1).

*2) The data in Figure 1B and C appear to establish that: 1. the channel protein SK2 can interact with MPP2 when the two proteins are expressed in HEK cells, and 2. SK2 does not interact with PSD-95 when the two proteins are co-expressed in HEK cells. However, the data would be more convincing if the authors indicated the molecular weight of the immunoprecipitated MPP2 band, and showed an entire lane, identifying any other cross-reacting bands.*

Done, as requested; new Figure 2.

*In the second paragraph of the subsection “MPP2 interacts with SK2 “the authors indicate a control as "data not shown." These are important controls and the data should be presented.*

Done, as requested, please see new Figure 2—figure supplement 1 and Figure 2—figure supplement 2.

*3) The data in Figure 1 presented problems for two reviewers:*

*A) The data in Figure 1 shows that the SK2L isoform does not interact with Shank, SAP97, or CACNB4 when co-expressed; however, this data does not exclude interaction of SK2-S with SAP-97.*

To address this directly, we co-expressed C8-SAP97 with SK2-S and performed immunoprecipitation using anti-SK2 antibody. Probing a Western blot of the immunoprecipitated proteins for C8-SAP97 did not detect a band, although C8-SAP97 expression was verified by probing the input with C8 antibody. As a further control we co-expressed C8-SAP97 and GluA1, that are known to interact; immunoprecipitation using anti-GluA1 antibody, co-precipitated C8-SAP97. These data are shown in new Figure 2—figure supplement 1.

*In the subsection “MPP2 interacts with SK2”, referring to Figure 1, the authors state that "*…*several other MAGUK proteins (Shank1, SAP97, CaCNB4) were prepared*…

*and used as baits for the His-tagged N-terminal domain of SK2-L." Shank1 and CaCNB4 are not MAGUK proteins and do not have an SH3-HOOK-GK domain. The experiment in Figure 1 shows that the SH3-HOOK-GK domain of SAP-97 doesn't pull down SK2-L. The significance of its lack of interaction with Shank1 and CaCNB4 is less clear.*

CaCNB4 is a non-canonical MAGUK protein lacking N-terminal PDZ domains but having SH3-HOOK-GK domains (Chen et al., Nature, 2004; Van Petegam et al., Nature, 2004; http://channelpedia.epfl.ch/ionchannels/91). SHANK is not a MAGUK protein but is a synaptic scaffold with an SH3 domain. For clarity, we have eliminated SHANK1 from the revised manuscript and focus, in new Figure 2 on SAP97 and CaCNB4, two proteins with SH3-HOOK-GK domains that are in different subfamilies from MPP2.

*B) A major part of the text that refers to Figure 1 (subsection “MPP2 interacts with SK2”, second paragraph) describes experiments showing that the SH3-HOOK-GK domain of MPP2 is the domain that is required for interaction with SK2-L. While Figure 1 may perhaps represent the data for this, there is no reference to Figure 1 in the text. In addition, the legend for Figure 1 makes no mention of SH3-HOOK-GK domains (but does refer to GST-pulldowns) and gives the impression that the full length proteins are being used here. The Methods do not give any information that allows one to identify what sequence is being used as bait in these experiments. Finally, the text erroneously refers to Figure 1 as showing that Kv1.4 pulls down PSD-93. In summary, either an important part of the text is missing or these experiments have to be described more clearly (or both).*

This section has been thoroughly reworded and appropriate details are now included.

*4) The immunohistochemistry and EM analysis that represent the core of MPP2 analysis are well done but need to be accompanied by intermediate resolution staining that shows the subcellular location of MPP2 in synapses of cultured neurons and whether or not the protein exhibits a distinct pattern of neuronal distribution from the marker protein PSD-95.*

We are not sure why the reviewers ask for lower resolution staining in a preparation, neuronal cultures, that is further removed from the native condition, since we show clearly much higher resolution data to determine the subcellular localization of MPP2 in intact brain tissue, indeed at the level of individual synaptic membranes. If cultured neurons have synapses mirroring native synapses, we would see immunoreactivity for MPP2 – or for PSD-95 – in spines without resolution to the synapse itself that is only provided by post-embedding immunoEM. It is also unclear what additional information would be conveyed by comparisons to PSD-95. But in fact we have studied PSD-95 localization by iEM in previous work (Lin et al., Nat. Neurosci. 2008; Allen et al., Nat. Neurosci. 2011; Fukaya and Watanabe, J. Comp. Neurol. 2000), and we see no difference in the subcellular localization of MPP2 and PSD-95. Both proteins are integral PSD components and are also localized to a lesser extent to the dendrites. Consistent with other studies with PSD-95, MPP2 also appears to be distributed evenly across the PSD consistent with comprising part of a backbone that stabilizes its binding partners at the synapse (see DeGioris, Galbraith, Dosemeci, Chen and Reese, Brain, Cell Biol.2006)

5) The data in Figure 4 is not a convincing demonstration of specific knockdown of MPP2. The authors should demonstrate the specificity of knockdown in cultured neurons where more quantitative measures are possible and where the possibility of off-target effects can be assessed. For example, is the expression of SAP97 altered?

We respectfully disagree with the reviewers on this point. The data in original Figure 4 (now, Figure 5) are quantified at very high resolution by comparing transfected and non-transfected spines and dendrites at the level of electron microscopy, and quite clearly demonstrate effective knockdown. We have analyzed particle distribution by ANOVA and added the appropriate p value (< 0.001).

The reviewers request quantitation of knockdown in cultured neurons, but they present problems that are circumvented by the higher resolution quantitation of knockdown we have provided. In cultures, knockdown analysis would be measured by transfecting them with shRNA plasmids and subsequent Western blots. Since cultured neurons are difficult to transfect, quantifying efficiency of knockdown would be far less precise than the EM level quantification presented in original Figure 4 (new Figure 5).

Also, while neuronal cultures have been well used to study some aspects of synaptic physiology, particularly AMPA and NMDA receptor mediated events and trafficking, they do not fully reflect native synapses. For example we have never recorded an apamin effect on EPSPs, nor even an SK current in cultured neurons. Thus, while cultured neurons may be useful for insights into glutamate receptors, it has become clear in the last several years that additional conductances make important contributions to post-synaptic responses in native spines and not all of these are recapitulated in neuronal cultures.

We very much appreciate the possibility of off-target effects and for this reason we applied the most direct and potent measure of knockdown specificity: re-establishment of the wild type phenotype by expression of an shRNA immune MPP2. It is indeed possible that SAP97 levels, or other synaptic proteins are reduced or altered by MPP2 shRNA expression. Using our online tool, we found no evidence that SAP97 mRNA would be knocked down by the MPP2 shRNAs. Thus if MPP2 knockdown is associated with altered levels of SAP97 this is more likely to be a consequence of MPP2 knockdown and compensatory changes at the level of the synapse than to a spurious cross-reaction of the shRNA with SAP97 mRNA. Our follow up studies are focused on collateral consequences of MPP2 loss for synaptic physiology. It is unclear how far the reviewers want us to go to search for possible off-target effects when we provide the explicit specificity control of the shRNA immune MPP2 rescuing the knockdown phenotype on which this work is focused.